# On the Convergence of Smooth Regularized Approximate Value Iteration Schemes

**Elena Smirnova**
esmirnovae@gmail.com

**Elvis Dohmatob**
Criteo AI Lab
e.dohmatob@criteo.com

## Abstract

Entropy regularization, smoothing of Q-values and neural network function approximator are key components of the state-of-the-art reinforcement learning (RL) algorithms, such as Soft Actor-Critic [1]. Despite the widespread use, the impact of these core techniques on the convergence of RL algorithms is not yet fully understood. In this work, we analyse these techniques from error propagation perspective using the approximate dynamic programming framework. In particular, our analysis shows that (1) value smoothing results in increased stability of the algorithm in exchange for slower convergence, (2) entropy regularization reduces overestimation errors at the cost of modifying the original problem, (3) we study a combination of these techniques that describes the Soft Actor-Critic algorithm.

## 1 Introduction

In practical settings, the reinforcement learning (RL) algorithms are faced with a challenge of maximizing the cumulative reward given a finite sample of environment transitions and inexact representation of policy and value function. This gives rise to errors that propagate across learning iterations and, combined, can result in divergence. Recently, state-of-the-art RL algorithms have been successful in solving complex environments and, hence, overcoming inaccuracies and their accumulation.

A number of techniques is commonly used in the large-scale RL setting, namely, entropy regularization, smoothing of Q-values and neural network function approximation. The *entropy regularized* methods [2, 3, 1] have shown robust behaviour on a variety of simulated and real-world tasks. In addition to regularization, *smoothing of Q-value function*, known as "soft" or delayed target update, is reported to increase stability of the algorithm and lower the risk of divergence [4, 5, 1]. Besides entropy regularization and value smoothing, notable progress on complex environments has been achieved with the use of *neural network function approximators* [6].

Despite practically valuable results, the theory behind these components is incomplete. Recently, several theoretical frameworks have been proposed to capture a multitude of entropy-regularized methods [7, 8, 9], still the practical effectiveness of entropy regularization is not yet fully understood. Recent works contributed to the theoretical grounding of the state-of-the-art entropy regularized policy-gradient algorithms [10, 11] and understanding the geometry of their objective function [12].

**Contributions.** In this paper, we show the implications of the above techniques on error propagation and convergence of actor-critic and value-based RL algorithms. We carry the error propagation analysis of abstract algorithms implementing entropy regularization and value smoothing using approximate dynamic-programming framework [13].

In detail, (1) We formalize the value smoothing technique using a new smooth Bellman operator. The analysis of the resulting dynamic programming scheme shows that value smoothing results in increased stability in exchange for slower convergence. (2) Based on the formalism of regularized Bellman operators [8], we show that the entropy regularization reduces overestimation errors [14] at the cost of modifying the original problem. The amount of this reduction depends on the size of state-action advantages scaled by the regularization parameter. (3) Motivated by the popular

Soft Actor-Critic algorithm [1], we provide error bounds for an abstract algorithm that combines smoothing with regularization and utilises neural network function approximation.

**Related work**   Prior works build their analysis on the top of approximate dynamic programming framework [13] (ADP). Regularized ADP [8] unifies (relative) entropy regularized algorithms through the use of regularization function. A related value-based scheme [9] generalizes the entropy regularized value iteration and gap-increasing methods. Another extension of ADP [15] proposes a value iteration algorithm with time-varied degree of value smoothing. Yet another study [16] shows that the KL divergence regularizer leads to the error averaging effect. Our work is different from the above-cited work since (1) we study a fixed value smoothing through a new type of Bellman operator, (2) we provide an alternative analysis of the regularized value iteration that highlights the effectiveness of entropy regularization.

**Outline**   The paper is organized as follows. First, the necessary background is presented in Sec. 2. Next, we analyse the value smoothing technique in Sec 3 and the entropy regularization in Sec. 4. Finally, in Sec. 5 we introduce an abstract algorithm that combines both techniques and neural network function approximation as in the Soft Actor-Critic algorithm.

## 2   Setting

### 2.1   Markov decision process

$\Delta_X$ will denote the set of probability distributions over finite set (or general measurable space) $X$ and $Y^X$ is a set of functions from set $X$ to set $Y$. We consider a Markov decision process (MDP) is a tuple $M := (\mathcal{S}, \mathcal{A}, P, r, \gamma)$ where $\mathcal{S}$ is a state space, $\mathcal{A}$ is a finite action space, $P \in \Delta_{\mathcal{S}}^{\mathcal{S} \times \mathcal{A}}$ is the transition kernel with $P(s'|s, a)$ being a probability of environment moving to a state $s'$ given an action $a$ in a state $s$, accompanied by a reward $r(s, a)$ (assumed to be bounded). We define a stochastic stationary policy $\pi \in \Delta_{\mathcal{A}}^{\mathcal{S}}$. We consider the discounted setting with discount factor $\gamma \in [0, 1)$.

We define the Bellman operator $\mathcal{T}^{\pi}$ for any function $V \in \mathbb{R}^{\mathcal{S}}$, $\forall s \in \mathcal{S}$ as follows

$$[\mathcal{T}^{\pi} V](s) := \mathbb{E}_{a \sim \pi(\cdot|s)} \left[ r(s, a) + \gamma \mathbb{E}_{s' \sim P(\cdot|s, a)}[V(s')] \right] = r^{\pi}(s) + \gamma P^{\pi}(\cdot|s) V, \tag{1}$$

where $r^{\pi} \in \mathbb{R}^{\mathcal{S}}$ and $P^{\pi} \in \Delta_{\mathcal{S}}^{\mathcal{S}}$ are defined by $r^{\pi}(s) := \mathbb{E}_{a \sim \pi(\cdot|s)}[r(s, a)]$ and $P^{\pi}(s'|s) := \mathbb{E}_{a \sim \pi(\cdot|s)}[P(s'|s, a)]$. $\mathcal{T}^{\pi}$ is a $\gamma$-contraction in $\ell_{\infty}$ norm and its unique fixed-point is $V^{\pi} :=$ $\lim_{k \to \infty} (\mathcal{T}^{\pi})^k V = V^{\pi}$, where equality holds component-wise. For any function $V \in \mathbb{R}^{\mathcal{S}}$, associate a function $Q_V \in \mathbb{R}^{\mathcal{S} \times \mathcal{A}}$ as follows $Q_V(s, a) := r(s, a) + \gamma \mathbb{E}_{s' \sim P(s'|s, a)}[V(s')]$.

Finally, we define the optimal Bellman operator as follows (the max is point-wise)

$$\mathcal{T} V := \max_{\pi} \mathcal{T}^{\pi} V, \tag{2}$$

which again is a $\gamma$-contraction in $\ell_{\infty}$ norm and its unique fixed-point is the optimal value function $V^*$. We denote by $\mathcal{G}(V)$ the set of optimal (greedy) policies that achieve the maximum of (2) state-wise

$$\mathcal{G}(V) := \arg \max_{\pi} \mathcal{T}^{\pi} V \subseteq \Delta_{\mathcal{A}}^{\mathcal{S}}.$$

Further, we will omit the per-state notation and imply that the relations hold state-wise.

### 2.2   Modified policy-iteration schemes

**Modified Policy Iteration (MPI) [17].**   MPI is a classical dynamic programming algorithm that alternates between policy improvement and (partial) policy evaluation steps. For $m \geq 1$, the $m$-step MPI algorithm is defined as follows

$$\text{(MPI)} \qquad \begin{cases} \pi_{t+1} \leftarrow \mathcal{G}(V_t) \\ V_{t+1} \leftarrow (\mathcal{T}^{\pi_{t+1}})^m V_t, \end{cases} \tag{3}$$

where $m = 1$ corresponds to Value Iteration and $m = \infty$ corresponds to Policy Iteration. Here $V_t$ denotes an approximation of $V^{\pi_t}$.

**Approximate Modified Policy Iteration (AMPI) [13].** AMPI is an approximate counterpart of (3) that can be seen as a generalization of MPI that allows errors in the policy improvement ($\epsilon'_t$) and policy evaluation ($\epsilon_t$) steps

$$\text{(AMPI)} \qquad \begin{cases} \pi_{t+1} \leftarrow \mathcal{G}^{\epsilon'_{t+1}}(V_t) \\ V_{t+1} \leftarrow (\mathcal{T}^{\pi_{t+1}})^m V_t + \epsilon_{t+1}, \end{cases} \tag{4}$$

where $\epsilon_t, \epsilon'_t \in \mathbb{R}^{\mathcal{S}}$ are respectively the evaluation step and the policy improvement step error vectors (one component per state) and $\pi \in \mathcal{G}^{\epsilon'}(V) \iff \forall \pi' \ \mathcal{T}^{\pi'} V \le \mathcal{T}^{\pi} V + \epsilon'$. AMPI naturally arises from MPI in practical settings with large or infinite state-action spaces. Commonly, the errors include sampling errors due to the limited number of rollouts at each iteration and function approximation errors due to solving the regression problem to represent policy and value function. From (4), the approximation errors are defined as follows

$$\epsilon_t := V_t - \mathcal{T}^{\pi_t} V_{t-1}, \quad \epsilon'_t := \mathcal{T} V_{t-1} - \mathcal{T}^{\pi_t} V_{t-1}. \tag{5}$$

The error propagation analysis links the error sequence that occurred at previous iterations to the distance to optimality of the current value iterate. The following Lemma shows that (a) the convergence of the AMPI algorithm depends on the cumulative error term $E_N$, (b) the error-free MPI algorithm converges with the rate $\gamma$.

**Lemma 1** (AMPI error propagation [13]). *For any initial value function $V_0$, consider the AMPI scheme* (4) *with $m \ge 1$. Then, one has*

$$\|V^{\pi_N} - V^*\|_\infty \le \frac{2}{1-\gamma} \left( E_N + \gamma^N \|V_0 - V^*\|_\infty \right), \tag{6}$$

*where $E_N := \sum_{t=1}^{N-1} \gamma^{N-t}(\|\epsilon_t\|_\infty + \|\epsilon'_t\|_\infty)$.*

## 3 Smooth Approximate Modified Policy Iteration

Besides approximation, state-of-the-art RL algorithms employ value smoothing [4, 5] that consists in applying exponential smoothing rule to the result of approximate policy evaluation step $V_{t+1}$

$$\tilde{V}_{t+1} = (1 - \beta)\tilde{V}_t + \beta V_{t+1},$$

where $\beta \in (0, 1]$ controls the degree of smoothing. Value smoothing has been consistently reported to bring stability in the large-scale RL setting [4, 5, 1]. However, these practical benefits are not fully supported by theoretical understanding. In this paper, we analyse the smoothing technique using a new type of Bellman operator, called smooth Bellman operator.

### 3.1 Smooth Bellman operators

**Definition 1** (Smooth Bellman operators). *Given $\beta \in (0, 1]$ define a smooth Bellman operator $\mathcal{T}^{\pi}_{\beta} : V \to V$ as a convex combination of standard Bellman operator $\mathcal{T}^{\pi} : V \to V$ and identity operator $I : V \to V$*

$$\mathcal{T}^{\pi}_{\beta} := \beta \mathcal{T}^{\pi} + (1 - \beta)I, \tag{7}$$

*the optimal smooth Bellman operator and the set of smooth optimal policies*

$$\mathcal{T}_{\beta} := \max_{\pi} \mathcal{T}^{\pi}_{\beta}, \qquad \mathcal{G}_{\beta}(V) := \arg\max_{\pi} \mathcal{T}^{\pi}_{\beta} V.$$

It can be shown that smooth Bellman operators satisfy the contraction, monotonicity and distributivity properties (see proof in Appendix B.1). In particular, the (optimal) smooth Bellman operator is a contraction mapping with coefficient $\beta\gamma + (1 - \beta)$ in sup-norm

$$\|\mathcal{T}^{\pi}_{\beta} V_1 - \mathcal{T}^{\pi}_{\beta} V_2\|_\infty \le (\beta\gamma + (1 - \beta))\|V_1 - V_2\|_\infty.$$

This contraction property is weaker than for the standard Bellman operator since $0 \le \gamma \le \beta\gamma + (1 - \beta) < 1$ for $\gamma, \beta \in (0, 1]$. Importantly, the set of optimal policies of the smooth Bellman operator *remains the same* as with the one of the standard Bellman operator (see section 3.2 for more on this)

$$\mathcal{G}_{\beta}(V) := \arg\max_{\pi} \mathcal{T}^{\pi}_{\beta} V = \arg\max_{\pi} \mathcal{T}^{\pi} V,$$

and the optimal value function $V^*$ *remains the fixed point* of the optimal smooth Bellman operator

$$\mathcal{T}_\beta V^* = \beta \max_\pi \mathcal{T}^\pi V^* + (1-\beta) V^* = V^*.$$

## 3.2  Smooth AMPI scheme

Using the smooth Bellman operators, the approximate policy evaluation step and the smoothed update can be represented as follows by re-arranging terms

$$V_{t+1} = (1-\beta)V_t + \beta(r^\pi + \gamma P^\pi V_t + \epsilon_t) = \mathcal{T}_\beta^\pi V_t + \beta\epsilon_t.$$

Further, the set of $\beta$-smoothed $\beta\epsilon'$-approximation optimal policies equals to the standard set of $\epsilon'$-approximate optimal policies, i.e $\mathcal{G}^{\epsilon'}(V) = \mathcal{G}_\beta^{\beta\epsilon'}(V)$. Indeed,

$$
\begin{aligned}
\pi \in \mathcal{G}^{\epsilon'_t}(V_t) \; \forall \pi' &\iff \mathcal{T}^{\pi'} V_t \; \forall \pi' \leq \mathcal{T}^\pi V_t + \epsilon'_t \; \forall \pi' \\
&\iff \beta \mathcal{T}^{\pi'} V_t + (1-\beta)V_t \leq \beta \mathcal{T}^\pi V_t + (1-\beta)V_t + \beta\epsilon'_t \; \forall \pi' \\
&\iff \pi \in \mathcal{G}_\beta^{\beta\epsilon'_t}.
\end{aligned}
$$

This allows to formulate the value smoothing as a valid AMPI scheme with the smooth Bellman operator, called the smooth AMPI scheme

$$
\text{(smooth AMPI)} \quad
\begin{cases}
\pi_{t+1} = \mathcal{G}_\beta^{\beta\epsilon'_{t+1}}(V_t) \\
V_{t+1} = \mathcal{T}_\beta^{\pi_{t+1}} V_t + \beta\epsilon_{t+1}.
\end{cases}
\tag{8}
$$

where $\epsilon_t, \epsilon'_t \in \mathbb{R}^\mathcal{S}$ are the error vectors in the AMPI scheme as defined by (5). A direct consequence of the smooth AMPI scheme is that the size of approximation errors is lowered by a factor of $\beta$. Below, we detail the implications of smoothing on convergence through the errors propagation analysis of the scheme (8).

## 3.3  Error propagation of the smooth AMPI

**Proposition 1** (Smooth AMPI error propagation). *For any initial value function $V_0$, consider the smooth AMPI scheme* (8) *with smoothing coefficient $\beta \in (0,1]$. Let $\tilde\gamma := \beta\gamma + (1-\beta)$. Then*

$$\|V^{\pi_N} - V^*\|_\infty \leq \frac{2}{1-\tilde\gamma}\left(\beta E_N + \tilde\gamma^N \|V_0 - V^*\|_\infty\right),\tag{9}$$

*where $E_N := \sum_{t=1}^{N-1} \tilde\gamma^{N-t}(\|\epsilon_t\|_\infty + \|\epsilon'_t\|_\infty)$ and $\epsilon_t, \epsilon'_t \in \mathbb{R}^\mathcal{S}$ are the error vectors of approximate policy evaluation and policy improvement step defined by* (5).

**Increased stability**   The above error bound stabilizes training of the smooth AMPI compared to the standard AMPI (6) since the size of the error term $E_N$ is reduced by a smoothing factor of $\beta \in (0,1]$. The error down-weighting results in a smaller variance of performance and thus, increased stability. With sufficiently small $\beta$, the smooth AMPI induces lower risk of divergence, that arises due to the composition of approximation and contraction, i.e. $\|\mathcal{T}_\beta^\pi V_1 + \beta\epsilon_1 - \mathcal{T}_\beta^\pi V_2 - \beta\epsilon_2\|_\infty \leq \tilde\gamma\|V_1 - V_2\|_\infty + \beta\|\epsilon_1 - \epsilon_2\|_\infty < \|V_1 - V_2\|_\infty$ for $\beta \ll 1$. These benefits have been also observed empirically [1, 5, 4].

**Slower convergence**   The above benefits are achieved at the cost of slowing down the convergence through an increased value of the contraction coefficient $\tilde\gamma$, since the standard AMPI scheme convergences faster at rate $\gamma \leq \tilde\gamma$. Moreover, at a fixed number of iterations, the smooth AMPI scheme is more sensitive to initialization, e.g. random seed. Indeed, high variability in performance depending on a random seed is reported in the deep RL literature [5, 18].

**Numerical illustration**   We numerically confirm the implications of the smoothing technique on the convergence, provided by Proposition 4. We run experiments on a toy stochastic gridworld problem with the evaluation step error due to the sampling of state-transitions. We plot the performance loss over 30 runs with varying values of smoothing factor $\beta$. As can be seen from Figure 1, smaller values of $\beta$ result in tighter confidence intervals, but slower convergence speed.

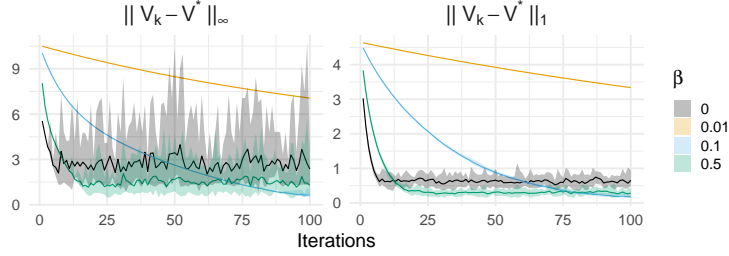

Figure 1: Performance loss computed over 30 runs of the smooth AMPI (8) with sampling of environment transitions under varying smoothing degree $\beta$. Larger values of $\beta$ induce smaller variance of performance, and hence, greater stability, at the cost of slower convergence.

**Remark.** *In practice, the smoothing is applied to the value network weights instead of outputs. Our analysis is still relevant in this case since (randomly initialized) value network with sufficiently large width and gradient descent training can be well-approximated by a linear model [19] (see Sec. 5.3).*

## 4  Entropy-regularized Approximate Value Iteration

Entropy regularization consists in modifying the standard RL objective to additionally maximize a per-state (relative) entropy of a policy. Regularized MPI framework summarizes the entropy regularized algorithms using the notion of regularized Bellman operator [8].

In this work, we focus on the value iteration instance of the regularized MPI, motivated by the practical algorithms, such as Soft Actor-Critic [1] and Soft Q-learning [3]. As in these algorithms, we consider the regularized Bellman operator with regularization function given by the negative entropy and a time-varying regularization parameter.

**Entropy regularized Bellman operators**  Define a strongly convex regularization function $\Omega : \Delta_{\mathcal{A}} \to \mathbb{R}$ given by a per-state negative entropy a policy

$$\Omega(\pi(\cdot|s)) = -\mathcal{H}(\pi(\cdot|s)) = \sum_a \pi(a|s) \log \pi(a|s).$$

Similarly to standard Bellman operators (1), we define the entropy regularized Bellman operator [8] with time-varying regularization parameter $\alpha_t > 0$

$$\mathcal{T}_{\Omega_t}^\pi V := \mathcal{T}^\pi V - \alpha_t \Omega(\pi), \tag{10}$$

By Fenchel duality, the optimal entropy regularized Bellman operator is a smooth maximum

$$\mathcal{T}_{\Omega_t} V := \max_{\pi \in \Delta_{\mathcal{A}}^{\mathcal{S}}} \mathcal{T}_{\Omega_t}^\pi V = (\Omega_t^*(Q_V(s, \cdot))_{s \in \mathcal{S}} = (\alpha_t \log \sum_a \exp(Q_V(s, a)/\alpha_t))_{s \in \mathcal{S}},$$

and the corresponding optimal policy $\mathcal{G}_{\Omega_t}(V) \in \Delta_{\mathcal{A}}^{\mathcal{S}}$ is given by the Boltzmann policy

$$(\pi_V^{\mathcal{B}}(\cdot|s))_{s \in \mathcal{S}} = \mathcal{G}_{\Omega_t}(V) := \arg\max_{\pi \in \Delta_{\mathcal{A}}^{\mathcal{S}}} \mathcal{T}_{\Omega_t}^\pi V = (\nabla \Omega_t^*(Q_V(s, \cdot))_{s \in \mathcal{S}},$$

that takes the form

$$\pi_V^{\mathcal{B}}(a|s) = \frac{\exp(Q_V(s, a)/\alpha_t)}{\sum_{a'} \exp(Q_V(s, a')/\alpha_t)}, \tag{11}$$

where $\alpha_t > 0$ is referred to as temperature parameter.

**Regularization gap**  Our error analysis makes use of the difference between regularized and standard optimal Bellman operators, called regularization gap.

**Definition 2** (Regularization gap). *For regularization function $\Omega : \Delta_{\mathcal{A}} \to \mathbb{R}$, define the regularization gap $\Omega^*(A_V(s, \cdot))$ at state $s \in \mathcal{S}$ as follows*

$$\Omega^*(A_V(s, \cdot)) := [\mathcal{T}_{\Omega_t} V](s) - [\mathcal{T} V](s)$$
$$= \Omega^*(Q_V(s, \cdot)) - \max Q_V(s, \cdot) = \Omega^*(Q_V(s, \cdot) - \max Q_V(s, \cdot))$$

*where $A_V(s, a) := Q_V(s, a) - \max Q_V(s, \cdot)$ is known as action advantage function or action gap.*

In the case of negative entropy regularizer, the regularization gap takes the form of smooth maximum over the state-action advantages, bounded as follows (see proof in Appendix C)

$$0 \le \Omega_t^*(A_V(s, \cdot)) := \alpha_t \log \sum_a \exp(A_V(s, a)/\alpha_t) \le \alpha_t \mathcal{H}(\pi_V^{\mathcal{B}}). \tag{12}$$

**Regularized value function**   Basic properties of Bellman operators including the $\gamma$-contraction are preserved in the case of regularized Bellman operators. It is well-known that the regularized algorithms return a value function different from the optimal value function $V^*$ [8]. For time-varying regularized algorithms, the regularized value function $V_N := \mathcal{T}_{\Omega_N}(\mathcal{T}_{\Omega_{N-1}}(\dots(\mathcal{T}_{\Omega_1} V_0)))$ is within a controllable distance to the optimal value function of the original problem.

**Lemma 2** (Regularized value function). *Consider time-varying regularization function $\Omega_t(\pi) := \alpha_t \Omega(\pi)$ for $\alpha_t > 0$. Denote the regularized value function after $N$ iterations $V_N := \mathcal{T}_{\Omega_N}(\mathcal{T}_{\Omega_{N-1}}(\dots(\mathcal{T}_{\Omega_1} V_0)))$. Then, $V_N$ satisfies*

$$\|V_N - V^*\|_\infty \le \sum_{t=1}^{N} \gamma^{N-t} \|\Omega_t^*(A_{V_{t-1}})\|_\infty + \gamma^N \|V_0 - V^*\|_\infty. \tag{13}$$

**Entropy-regularized approximate value iteration (reg-AVI)**   Consider approximate value iteration (AVI) algorithm that is an instance of the AMPI scheme with $m = 1$

$$(\text{AVI}) \qquad V_{t+1} \leftarrow \mathcal{T} V_t + \epsilon_{t+1}, \tag{14}$$

where $\epsilon_t \in \mathbb{R}^{\mathcal{S}}$ is the error vector of approximate update. AVI can be equivalently represented using the entropy regularized Bellman operators

$$(\text{reg-AVI}) \qquad V_{t+1} \leftarrow \mathcal{T}_{\Omega_t} V_t + \bar{\epsilon}_{t+1}, \tag{15}$$

where $\bar{\epsilon}_t \in \mathbb{R}^{\mathcal{S}}$ is the reg-AVI error vector, defined as a transformation of approximation errors

$$\bar{\epsilon}_{t+1} := V_{t+1} - \mathcal{T}_{\Omega_t} V_t = V_{t+1} - \mathcal{T}_{\Omega_t} V_t + (\mathcal{T} V_t - \mathcal{T} V_t) = V_{t+1} - \mathcal{T} V_t - \Omega_t^*(A_{V_t}) = \epsilon_{t+1} - \Omega_t^*(A_{V_t}).$$
$$\tag{16}$$

This equivalence suggests that the error bound for the reg-AVI is the same as for the AVI. This conclusion has also been obtained by [8] through a different analysis. Despite this equivalence, the reg-AVI remains beneficial as the regularized Bellman operator preserves the $\gamma$-rate of convergence, whereas the positive errors (overestimation) can be decreased (see Sec. 4.1). This improves the stability of the algorithm and lowers the risk of divergence.

## 4.1   Error propagation of the reg-AVI

**Proposition 2** (reg-AVI error propagation). *For any initial value function $V_0$, consider the entropy-based reg-AVI scheme (15) with time-varying regularization parameter $(\alpha_t)_t > 0$. Then, it holds that*

$$\|V_N - V^*\|_\infty \le E_N + A_N + \gamma^N \|V_0 - V^*\|_\infty, \tag{17}$$

*where $E_N := \sum_{t=1}^{N} \gamma^{N-t} \|\epsilon_t - \Omega_t^*(A_{V_{t-1}})\|_\infty$, $A_N := \sum_{t=1}^{N} \gamma^{N-t} \|\Omega_t^*(A_{V_{t-1}})\|_\infty$, the regularization gap $\Omega^*(A_V) := \mathcal{T}_\Omega V - \mathcal{T} V$ and the approximation errors of value update $\epsilon_t := V_t - \mathcal{T} V_{t-1}$.*

**Overestimation errors**   Consider positive errors $\epsilon_t := V_t - \mathcal{T} V_{t-1} \ge 0$, known as overestimation errors in actor-critic and value-based RL algorithms [20, 14, 21, 5]. These errors arise due to the noisy estimates of state-action values that, through the maximization step, result in a positive bias. For example, the noise can be induced by the function approximator [20] or can represent an

environmental noise [14]. Formally, by re-writing the Bellman operator w.r.t. Q-values

$$\epsilon_{t+1}(s, a) = [\mathcal{T}\hat{Q}_t](s, a) - [\mathcal{T}Q_t](s, a) = \gamma \mathbb{E}_{s' \sim P(\cdot|s,a)}[\max_{a'} \hat{Q}_t(s', a') - \max_{a'} Q_t(s', a')]$$

$$= \gamma \mathbb{E}_{s' \sim P(\cdot|s,a)}[\mathbb{E}_{\hat{\epsilon}}[\max_{a'}(Q_t(s', a') + \hat{\epsilon}_t(s', a'))] - \max_{a'} Q_t(s', a')] \geq 0,$$

where the last inequality follows from Jensen's inequality by convexity of maximum function.

The advantage of the bound (17) is that the cumulative error term $E_N$ could be made smaller by appropriately tuning the temperature parameter $\alpha_t$ to get the regularization gap $\Omega_t^*(A_{V_{t-1}})$ match the overestimation error $\epsilon_t$. In exchange, the $A_N$ term remains and results in a different regularized fixed point.

**Robustness to noise**    By (12), the regularization gap $\Omega_t^*(A_V)$ is non-negative and controllable. Thus, it follows from the above error bound that the overestimation errors of AVI can be reduced by performing the reg-AVI with appropriate scaling of temperature parameter. The reduction is larger for larger mean and variance of advantage values, i.e. difference of Q-values between the maximizing action and all other actions. This reduction comes in exchange for convergence to a different value function, related to the optimal value function by (26). As the $\gamma$-rate of convergence of the algorithm is preserved, together with error reduction, the reg-AVI results in a stabilizing effect in a noisy setting.

Our findings are supported by previous work that reports robustness to stochastic noise of the Soft Q-learning [3] and related algorithms [9]. Prior work [22] have shown that a related to the regularized Bellman operator – a softmax operator – can reduce the overestimation bias. Moreover, our result suggests similarity with advantage learning [23, 24] that amplifies the action advantages of Q-iterates. In particular, the advantage learning operator reduces the positive (overestimation) errors since the advantage values are negative

$$Q_{t+1}(s, a) \leftarrow [\mathcal{T}_{\mathrm{AL}}Q_t](s, a) + \epsilon_t(s, a) = [\mathcal{T}Q_t](s, a) + \epsilon_t(s, a) + \alpha A_{Q_t}(s, a).$$

Reinforcing the action gap is known to be advantageous in the presence of errors and stochasticity [25], especially in environments operating at a fine time scale, such as robotic platforms and video games.

**Temperature schedule**    The error bound (17) motivates the time-varying per-state temperature parameter $\alpha_t$ that depends on the size of overestimation errors and the regularization gap. The appropriate temperature matches the overestimation errors to the stochasticity of the policy. Towards a dynamic temperature adjustment, one can upper bound the size of overestimation errors using the variance of Q-values [26, 3.3.2] and approximate the regularization gap using the scaled entropy of the current policy, see (12).

# 5   Smooth Regularized Approximate Value Iteration

Motivated by the Soft Actor-Critic algorithm, we analyse an abstract algorithmic scheme that combines the entropy regularization (Sec. 4) with the value smoothing (Sec. 3). In addition, we study the function approximation errors induced by the policy and value network as in the Soft Actor-Critic algorithm (Sec. 5.3).

## 5.1   Smooth regularized Bellman operators

**Definition 3.** *For a smoothing parameter $\beta \in [0, 1)$ and a regularization parameter $\alpha > 0$, define the smooth regularized Bellman operator as applying smoothing to the regularized Bellman operator*

$$\mathcal{T}_{\Omega,\beta}^{\pi} := \beta \mathcal{T}_{\Omega}^{\pi} + (1 - \beta)I. \tag{18}$$

Similarly to smooth Bellman operator (see Sec. 3.1), smooth regularized Bellman operator is a contraction mapping with coefficient $\tilde{\gamma} := \gamma\beta + (1 - \beta)$ and satisfies basic properties of Bellman operators. Closely following Sec. 3.1, one can define the optimal smooth regularized Bellman operator $\mathcal{T}_{\Omega,\beta}$ and corresponding set of greedy policies $\mathcal{G}_{\Omega,\beta}$. The basic properties of optimal operator $\mathcal{T}_{\Omega,\beta}$ follow in the same way as for the evaluation operator.

**Smooth regularized value function and smooth reg-AVI**    For time-varying regularized algorithms, the smooth regularized value function $V_N := \mathcal{T}_{\Omega_N,\beta}(\mathcal{T}_{\Omega_{N-1},\beta}(\ldots(\mathcal{T}_{\Omega_1,\beta}V)))$ is within a

controllable distance to the optimal value function of the original problem (see proof in Appendix D.1)

$$\|V_N - V^*\|_\infty \le \beta \sum_{t=1}^{N} \tilde{\gamma}^{N-t} \|\Omega_t^*(A_{V_{t-1}})\|_\infty + \tilde{\gamma}^N \|V_0 - V^*\|_\infty.$$

Based on smooth regularized Bellman operators, we formulate the smooth regularized AVI scheme

$$\text{(smooth reg-AVI)} \qquad V_{t+1} \leftarrow \mathcal{T}_{\Omega_t,\beta} V_t + \beta \bar{\epsilon}_{t+1}, \tag{19}$$

where the errors $\bar{\epsilon}_t$ are the regularized AVI errors are defined by (16).

## 5.2 Error propagation of the smooth reg-AVI

**Theorem 1** (Smooth reg-AVI error propagation). *For any initial value function $V_0$, consider the smooth regularized AVI scheme* (19) *with smoothing parameter $\beta \in [0, 1)$ and time-varying temperature parameter $(\alpha_t)_t > 0$. Denote $\tilde{\gamma} := \beta \gamma + (1 - \beta)$. Then, one has*

$$\|V_N - V^*\|_\infty \le \beta \tilde{E}_N + \beta \tilde{A}_N + \tilde{\gamma}^N \|V_0 - V^*\|_\infty, \tag{20}$$

*where $\tilde{E}_N := \sum_{t=1}^{N} \tilde{\gamma}^{N-t} \|\epsilon_t - \Omega_t^*(A_{V_{t-1}})\|_\infty$, $\tilde{A}_N := \sum_{t=1}^{N} \tilde{\gamma}^{N-t} \|\Omega_t^*(A_{V_{t-1}})\|_\infty$, the regularization gap $\Omega^*(A_V) := \mathcal{T}_\Omega V - \mathcal{T} V$ and the approximation errors of value update $\epsilon_t := V_t - \mathcal{T} V_{t-1}$.*

This combined bound is beneficial since (a) it downweights terms $\tilde{E}_N$ and $\tilde{A}_N$ by a factor of $\beta \in (0, 1]$ (see Sec. 3.3), and (b) it allows to adjust the temperature parameter $\alpha_t$ to match the noise level, making the term $\tilde{E}_N$ even smaller (see Sec. 4.1). As a consequence, combining smoothing and regularization in the Soft Actor-Critic algorithm (a) increases stability of the algorithm provided by the value smoothing, and (b) reduces the value overestimation with appropriate scaling of regularization.

Besides the overestimation errors considered so far, the errors $\epsilon_t$ also include the estimation errors due to the sampled-based Bellman update $\hat{\mathcal{T}}$ and the approximation errors due to the limited capacity of the function approximator with parameters $\theta$ that represents the value function

$$\|\epsilon_{t+1}\| = \|V_{\theta_{t+1}} - \hat{\mathcal{T}}_{\Omega,\beta} V_t\| \le \|V_{\theta_{t+1}} - \mathcal{T}_{\Omega,\beta} V_t\| + \|\mathcal{T}_{\Omega,\beta} V_t - \hat{\mathcal{T}}_{\Omega,\beta} V_t\| = \|\epsilon_{t+1}^a\| + \|\epsilon_{t+1}^e\|. \tag{21}$$

Next, we detail the approximation errors $\epsilon_t^a$ in the case of neural network function class, utilised in the large-scale RL setting in general and, in particular, in the Soft Actor-Critic algorithm. We leave the estimation errors $\epsilon_{t+1}^e$ out of scope of this work. We show that the use of neural network is advantageous since the approximation errors vanish, under certain conditions, at increased width of the network. In contrast, previously studied linear function approximators require providing good features to achieve small approximation error [27, 9.5].

## 5.3 Neural network function approximation errors

The Soft Actor-Critic algorithm [1] utilises neural network function approximator to represent policy and value function. Motivated by this algorithm, we analyse the smooth reg-AVI update (19) performed by solving a neural network regression problem using gradient descent.

Formally, we consider the value network $V_\theta : \mathcal{S} \to \mathbb{R}$ trained with the smooth reg-AVI update

$$\text{(smooth reg-AVI)} \quad \theta_{k+1} \leftarrow \arg\min_\theta \|V_\theta - \mathcal{T}_{\Omega_k,\beta} V_k\|_2^2, \ V_{k+1} \leftarrow V_{\theta_{k+1}} \tag{22}$$

We assume that the optimization of the above problems is performed using the continuous-time gradient descent. We leave the analysis of its discretized version out of scope of this paper.

Similar to [28], we define recursively a *randomly initialized $L$-layer fully connected neural network* $f_\theta : \mathbb{R}^{m_0} \to \mathbb{R}^{m_L}$. Namely, for layer $h = 1, \dots, L$

$$f_\theta^{(h)}(x) = \theta^{(h)} g^{(h-1)}(x), \quad g^{(h)}(x) = \sqrt{\frac{c_\sigma}{m_h}} \sigma(f_\theta^{(h)}(x)),$$

where $\theta^{(h)} \in \mathbb{R}^{m_h \times m_{h-1}}$ is a weight matrix of $h$-layer of size $m_h$, initialized as i.i.d. Gaussians $\mathcal{N}(0, 1)$, $\sigma : \mathbb{R} \to \mathbb{R}$ is an element-wise activation function, $\sqrt{c_\sigma/m_h}$ is a scaling factor that ensures

that the norm $g^{(h)}(x)$ for each $h \in [\![L]\!]$ is approximately preserved at initialization. Assume $g^{(0)} = x$ and $m_0$ is the size of the input. Finally, we assume that the state space is finite of a size $S = |\mathcal{S}|$.

**Smooth reg-AVI update**  We consider that the value function is approximated using a value neural network $V := V_\theta \in \mathbb{R}^\mathcal{S}$ through the minimization problem (22). Note that in practical implementations the Bellman operator is evaluated using a target value network, e.g. in the Soft Actor-Critic [1]. Thus, the predicted values in (22) are considered fixed w.r.t. the current value network.

Let the minimization of problem (22) be achieved through the continuous-time limit of *full-batch* gradient descent on vector of parameters $\theta$, namely

$$\dot{\theta}(t) = -(\nabla_\theta V_\theta)^T (V_\theta - \mathcal{T}_{\Omega,\beta} V_k)\big|_{\theta=\theta(t)}.$$

In terms of value function, we obtain the following flow dynamics

$$\frac{dV_\theta}{dt} = \nabla_\theta V_\theta \dot{\theta}(t) = -(\nabla_\theta V_\theta)(\nabla_\theta V_\theta)^T (V_\theta - \mathcal{T}_{\Omega,\beta} V_k)\big|_{\theta=\theta(t)} = -K(t)(V_{\theta(t)} - \mathcal{T}_{\Omega,\beta} V_k)\big|_{\theta=\theta(t)},$$

where $K(t) \in \mathbb{R}^{S \times S}$ is a positive semi-definite matrix defined as $(K(t))_{s,s'} = \frac{\partial V_\theta(s)}{\partial \theta}^T \frac{\partial V_\theta(s')}{\partial \theta}\big|_{\theta=\theta(t)}$. It has been shown in [19] that $K(t) = K + \mathcal{O}(m^{-1/2})$ holds almost surely for large $m$ over random initializations of the parameters of the neural network $V_\theta$, i.e using neural network with width $\min_{l=1}^L m_l := m$, where $m \longrightarrow \infty$, and $S/m \longrightarrow \alpha \in [0,1)$. The deterministic limiting kernel $K \in \mathbb{R}^{S \times S}$ is called the limiting *Neural Tangent Kernel (NTK)* [28] of the neural network $V_\theta$ and is given by

$$K(s, s') = \mathbb{E}_{\theta_0 \sim \text{init}} \left[ \frac{\partial V_\theta(s)}{\partial \theta}^T \frac{\partial V_\theta(s')}{\partial \theta}\bigg|_{\theta=\theta_0} \right]. \tag{23}$$

The limiting NTK is defined over the state space where the gradient of the output of a randomly initialized value network with respect to its weights plays the role of (non-linear) features. Below, we show that the limiting NTK characterizes the approximation errors of a wide value network.

**Theorem 2** (Linear convergence of value network errors). *For notational simplicity, let us denote the value network $V(t) := V_\theta\big|_{\theta=\theta(t)}$ with sufficiently large width and the Bellman update $b_{k+1} := \mathcal{T}_{\Omega,\beta} V_k$. Then, if the limiting NTK of the neural net $V_\theta$ is positive definite, i.e its smallest eigenvalue is positive $\lambda_{\min}(K) > 0$, then the following contraction holds almost surely over all initializations $\theta(0)$ of the neural network*

$$|u_j^T (V(t) - b_{k+1})| \le \exp(-\lambda_{\min}(K)t)|u_j^T (V(0) - b_{k+1})|, \ \forall t \ge 0, \ j \in [\![S]\!],$$

*where $u_1, \ldots, u_S$ are the eigenvectors of $K$. In particular, the process $V(t)$ converges (in any norm) to $b_{k+1}$ almost surely over random initializations of $V_\theta$ at linear rate $e^{-\lambda_{\min}(K)}$.*

**Corollary 1** (Value network errors of the smooth reg-AVI). *Under conditions of Theorem 2, the neural network approximation error of problem (22) optimized by gradient descent is upper bounded by*

$$\|\epsilon_{k+1}^a\| = \|V_{\theta_{k+1}} - \mathcal{T}_{\Omega,\beta} V_k\| \stackrel{a.s.}{=} \mathcal{O}(e^{-\lambda_{\min}(K)T}),$$

*where $K$ is the limiting NTK of the value network $V_\theta$ and $T$ is the number of gradient descent steps.*

Previously established bounds on approximation errors require computing the VC-dimension of a function class [13]. Our analysis is different in that it uses the limiting NTK to characterize the approximation errors of a wide neural network trained using gradient descent.

# 6 Concluding remarks

We have shown the implications of key components of the state-of-the-art RL algorithms, such as value smoothing, entropy regularization and neural network approximators, on convergence of actor-critic and value-based RL algorithms. We carried the error propagation analysis of abstract algorithms implementing entropy regularization and value smoothing using approximate dynamic-programming framework, and provided explicit bounds on the error to optimality.

## Broader impact

This research benefits the RL researchers and practitioners running large-scale RL applications. It improves theoretical understanding of the state-of-the-art techniques from error propagation perspective. This is helpful in algorithm design for a particular application. Our analysis builds on the top of approximate dynamic-programming framework and might not cover all the implications of the above-mentioned techniques.

## Acknowledgments and Disclosure of Funding

Revenues related to this work: Elvis Dohmatob is a full-time employee of Criteo AI Lab.

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
