[Supplementary Material]

# A AMPI with $m = 1$

Recall the approximate modified policy iteration (AMPI) algorithmic scheme (4) with $m = 1$

$$\begin{cases} \pi_{t+1} \in \mathcal{G}^{\epsilon'_{t+1}}(V_t) \\ V_{t+1} = \mathcal{T}^{\pi_{t+1}} V_t + \epsilon_{t+1}, \end{cases}$$

where $\epsilon_t, \epsilon'_t \in \mathbb{R}^{\mathcal{S}}$ are respectively the evaluation step and the policy improvement step error vectors (one component per state) and $\pi \in \mathcal{G}_{\epsilon'}(V) \iff \forall \pi' \; \mathcal{T}^{\pi'} V \leq \mathcal{T}^{\pi} V + \epsilon'$.

Next, we establish the error propagation bound for the above scheme, that we will use to obtain the one for the smooth AMPI.

## A.1 Error propagation

**Lemma 3** (AMPI error propagation). *For any initial value function $V_0$, consider the AMPI scheme (4) with $m = 1$. Then, one has*

$$\|V^{\pi_N} - V^*\|_\infty \leq \frac{2}{1 - \gamma} \left( E_N + \gamma^N \|V_0 - V^*\|_\infty \right), \tag{24}$$

*where $E_N := \sum_{t=1}^{N-1} \gamma^{N-t}(\|\epsilon_t\|_\infty + \|\epsilon'_t\|_\infty)$.*

*Proof.* By triangular inequality and $\gamma$-contraction property of Bellman operators, one has

$$\begin{aligned}
\|V^* - V_N\|_\infty &= \|\mathcal{T}V^* - \mathcal{T}^{\pi_N} V_{N-1} - \epsilon_N\|_\infty \\
&= \|\mathcal{T}V^* - \mathcal{T}V_{N-1} + \mathcal{T}V_{N-1} - \mathcal{T}^{\pi_N} V_{N-1} - \epsilon_N\|_\infty \\
&\leq \gamma\|V^* - V_{N-1}\|_\infty + \|\epsilon_N\|_\infty + \|\epsilon'_N\|_\infty \\
&= \sum_{t=1}^{N} \gamma^{N-t}(\|\epsilon_t\|_\infty + \|\epsilon'_t\|_\infty) + \gamma^N \|V^* - V_0\|_\infty,
\end{aligned}$$

where the last inequality is due to the definition of the greedy operator $\forall \pi' \; \mathcal{T}^{\pi'} V_{N-1} \leq \mathcal{T}^{\pi_N} V_{N-1} + \epsilon'_N$, including $\pi' = \arg\max_\pi \mathcal{T}^\pi V_{N-1}$.

The claim follows from the fact that $\|V^* - V^{\pi_N}\|_\infty \leq \frac{2\gamma}{1-\gamma}\|V^* - V_{N-1}\|_\infty$. Indeed, by triangular inequality and $\gamma$-contraction property of Bellman operators

$$\begin{aligned}
\|V^* - V^{\pi_N}\|_\infty &= \|\mathcal{T}V^* - \mathcal{T}^{\pi_N} V_{N-1} + \mathcal{T}^{\pi_N} V_{N-1} - \mathcal{T}^{\pi_N} V^{\pi_N}\|_\infty \\
&\leq \|\mathcal{T}V^* - \mathcal{T}V_{N-1}\|_\infty + \gamma\|V_{N-1} - V^{\pi_N}\|_\infty \\
&\leq \gamma\|V^* - V_{N-1}\|_\infty + \gamma(\|V_{N-1} - V^*\|_\infty + \|V^* - V^{\pi_N}\|_\infty) \\
&\leq 2\gamma \sum_{t=0}^{\infty} \gamma^t \|V^* - V_{N-1}\|_\infty \\
&= \frac{2\gamma}{1 - \gamma}\|V^* - V_{N-1}\|_\infty.
\end{aligned}$$

$\square$

# B Smooth AMPI with $m = 1$

We prove properties of the smooth Bellman operators in Section B.1. We obtain the error bound on the smooth AMPI with $m = 1$ in Section B.2.

## B.1 Smooth Bellman operators

**Proposition 3.** *Denote $\tilde{\gamma} := \beta\gamma + (1 - \beta)$. The smooth Bellman operators $\mathcal{T}^\pi_\beta, \mathcal{T}_\beta$ defined by (1) satisfy the following properties:*

1. *Monotonicity: for $V_1, V_2 \in \mathbb{R}^{\mathcal{S}}$ such that $V_1 \geq V_2$, one has*

$$\mathcal{T}_\beta^\pi V_1 \geq \mathcal{T}_\beta^\pi V_2, \quad \mathcal{T}_\beta V_1 \geq \mathcal{T}_\beta V_2.$$

2. *Distributivity: for $c \in \mathbb{R}$ and $V \in \mathbb{R}^{\mathcal{S}}$, one has*

$$\mathcal{T}_\beta^\pi (V + c\mathbf{1}) = \mathcal{T}_\beta^\pi V + \tilde{\gamma} c \mathbf{1}, \quad \mathcal{T}_\beta (V + c\mathbf{1}) = \mathcal{T}_\beta V + \tilde{\gamma} c \mathbf{1}.$$

3. *Contraction in sup-norm with coefficient $\tilde{\gamma}$: for $V_1, V_2 \in \mathbb{R}^{\mathcal{S}}$, one has*

$$\|\mathcal{T}_\beta^\pi V_1 - \mathcal{T}_\beta^\pi V_2\|_\infty \leq \tilde{\gamma}\|V_1 - V_2\|_\infty, \quad \|\mathcal{T}_\beta V_1 - \mathcal{T}_\beta V_2\|_\infty \leq \tilde{\gamma}\|V_1 - V_2\|_\infty.$$

*Proof.* We first prove the claimed properties for the smooth Bellman operator $\mathcal{T}_\beta^\pi$.

1. Monotonicity. Follows from monotonicity of the standard Bellman operator and the monotonisity of the identity function.

2. Distributivity. By distributivity of the standard Bellman operator $\mathcal{T}^\pi(V + c\mathbf{1}) = \mathcal{T}^\pi V + \gamma c \mathbf{1}$, one has

$$\begin{aligned}
\mathcal{T}_\beta^\pi (V + c\mathbf{1}) &= \beta \mathcal{T}^\pi (V + c\mathbf{1}) + (1 - \beta)(V + c\mathbf{1}) \\
&= \beta (\mathcal{T}^\pi V + \gamma c\mathbf{1}) + (1 - \beta)V + (1 - \beta)c\mathbf{1} \\
&= \mathcal{T}_\beta^\pi V + \tilde{\gamma} c \mathbf{1}.
\end{aligned}$$

3. Contraction. By contraction of the standard Bellman operator $\|\mathcal{T}^\pi V_1 - \mathcal{T}^\pi V_2\|_\infty \leq \gamma\|V_1 - V_2\|_\infty$, one has

$$\begin{aligned}
\|\mathcal{T}_\beta^\pi V_1 - \mathcal{T}_\beta^\pi V_2\|_\infty &= \|\beta \mathcal{T}^\pi V_1 + (1 - \beta)V_1 - \beta \mathcal{T}^\pi V_2 - (1 - \beta)V_2\|_\infty \\
&\leq \beta \|T^\pi V_1 - T^\pi V_2\|_\infty + (1 - \beta)\|V_1 - V_2\|_\infty \\
&\leq \beta \gamma \|V_1 - V_2\|_\infty + (1 - \beta)\|V_1 - V_2\|_\infty \\
&\leq \tilde{\gamma}\|V_1 - V_2\|_\infty.
\end{aligned}$$

For the optimal smooth Bellman operator $\mathcal{T}_\beta$, the proof is the same as above by considering properties of the optimal Bellman operator $\mathcal{T}$. $\qquad \square$

### B.2 Error propagation

**Proposition 4** (Smooth AMPI error propagation). *For any initial value function $V_0$, consider the smooth AMPI scheme (8) with smoothing coefficient $\beta \in (0, 1]$. Let $\tilde{\gamma} := \beta\gamma + (1 - \beta)$. Then*

$$\|V^{\pi_N} - V^*\|_\infty \leq \frac{2}{1 - \tilde{\gamma}} \left( \beta E_N + \tilde{\gamma}^N \|V_0 - V^*\|_\infty \right), \tag{25}$$

*where $E_N := \sum_{t=1}^{N-1} \tilde{\gamma}^{N-t}(\|\epsilon_t\|_\infty + \|\epsilon_t'\|_\infty)$ and $\epsilon_t, \epsilon_t' \in \mathbb{R}^{\mathcal{S}}$ are the error vectors of approximate policy evaluation and policy improvement step defined by (5).*

*Proof.* The result follows from Lemma 3 with contraction coefficient $\tilde{\gamma}$ and $\beta$-rescaled errors. $\quad \square$

## C   Regularized AVI

In the following, we consider a regularization function given by the (per-state) negative entropy of a policy

$$\Omega(\pi) := (-\alpha \mathcal{H}(\pi(\cdot|s)))_{s \in \mathcal{S}},$$

and the time-varying regularization function given a sequence of positive weights $(\alpha_t)_t > 0$

$$\Omega_t(\pi) := \alpha_t \Omega(\pi).$$

In addition, we focus on the value iteration instance of the AMPI.

We first prove an auxiliary result that describes the difference between the regularized and the standard optimal Bellman operators.

**Lemma 4** (Regularization gap). *For any value function $V \in \mathbb{R}^{\mathcal{S}}$, the regularization gap*

$$\Omega^*(A_V(s, \cdot)) := [\mathcal{T}_\Omega V](s) - [\mathcal{T}V](s)$$

*satisfies for all $s \in \mathcal{S}$*

$$0 \leq \Omega^*(A_V(s, \cdot)) := \alpha \log \sum_{a \in \mathcal{A}} \exp(A(s, a)/\alpha) \leq \alpha \mathcal{H}(\pi_V^{\mathcal{B}}(\cdot|s)),$$

*where $\pi_V^{\mathcal{B}}$ is a Boltzmann policy induced by the value function $V$ as defined by* (11).

*Proof.* By definition of the regularization gap and properties of smooth maximum, one has

$$\begin{aligned}
\Omega^*(A_V(s, \cdot)) &= [\mathcal{T}_\Omega V](s) - [\mathcal{T}V](s) \\
&= \Omega^*(Q_V(s, \cdot)) - \max Q_V(s, \cdot) \\
&= \Omega^*(Q_V(s, \cdot) - \max Q_V(s, \cdot)) \\
&= \alpha \log \sum_{a \in \mathcal{A}} \exp(A(s, a)/\alpha).
\end{aligned}$$

The lower bound is obtained using boundness of smooth maximum

$$0 \leq \mathcal{T}_\Omega V - \mathcal{T}V \iff 0 \leq \mathcal{T}^{\mathcal{G}_\Omega} V - \mathcal{T}^{\mathcal{G}} V + \alpha \mathcal{H}(\pi_V^{\mathcal{B}}).$$

From here, the upper bound follows due to the maximizing property of the greedy policy

$$-\alpha \mathcal{H}(\pi_V^{\mathcal{B}}) \leq \mathcal{T}^{\mathcal{G}_\Omega} V - \mathcal{T}^{\mathcal{G}} V \leq 0 \iff 0 \leq \mathcal{T}_\Omega^{\mathcal{G}_\Omega} V - \mathcal{T}^{\mathcal{G}} V \leq \alpha \mathcal{H}(\pi_V^{\mathcal{B}}).$$

$\square$

### C.1 Regularized value function

The regularization changes the original RL problem. The next lemma relates the regularized value function to the optimal value function in the original problem.

**Lemma 5** (Regularized value function). *Consider time-varying regularization function $\Omega_t(\pi) := \alpha_t \Omega(\pi)$ for $\alpha_t > 0$. Denote the regularized value function after $N$ iterations $V_N := \mathcal{T}_{\Omega_N}(\mathcal{T}_{\Omega_{N-1}}(\ldots(\mathcal{T}_{\Omega_1} V_0)))$. Then, $V_N$ satisfies*

$$\|V_N - V^*\|_\infty \leq \sum_{t=1}^{N} \gamma^{N-t} \|\Omega_t^*(A_{V_{t-1}})\|_\infty + \gamma^N \|V_0 - V^*\|_\infty. \tag{26}$$

*Proof.* By putting $V := V_0$, we first prove that

$$\mathcal{T}^N V \leq V_N \leq \mathcal{T}^N V + \sum_{t=1}^{N} \gamma^{N-t} \|\Omega_t^*(A_{V_{t-1}})\|_\infty \mathbf{1}.$$

To do so, note that $\mathcal{T}_\Omega \geq \mathcal{T}$ since $\mathcal{T}_{\Omega_t} V = \mathcal{T}V + \Omega_t^*(A_V)$ from Lemma 4. Then, the lower bound follows by induction from

$$\mathcal{T}_{\Omega_2}(\mathcal{T}_{\Omega_1} V) \geq \mathcal{T}_{\Omega_2}(\mathcal{T}V) \geq \mathcal{T}(\mathcal{T}V) = \mathcal{T}^2 V \Rightarrow V_N \geq \mathcal{T}^N V.$$

Similarly, the upper bound follows from

$$\begin{aligned}
\mathcal{T}_{\Omega_2}(\mathcal{T}_{\Omega_1} V) &= \mathcal{T}_{\Omega_2}(\mathcal{T}V + \Omega_t^*(A_V)) \\
&= \mathcal{T}(\mathcal{T}V + \Omega_t^*(A_V)) + \Omega_t^*(A_{V_1}) \\
&\leq \mathcal{T}(\mathcal{T}V + \|\Omega_t^*(A_V)\|_\infty \mathbf{1}) + \Omega_t^*(A_{V_1}) \\
&\leq \mathcal{T}^2 V + \gamma \|\Omega_t^*(A_V)\|_\infty \mathbf{1} + \Omega_t^*(A_{V_1}) \\
&\Rightarrow V_N \leq \mathcal{T}^N V + \sum_{t=1}^{N} \gamma^{N-t} \|\Omega_t^*(A_{V_{t-1}})\|_\infty \mathbf{1}.
\end{aligned}$$

The claimed result is obtained by noticing that

$$V^* - \gamma^N \|V^* - V\|_\infty \mathbf{1} \le \mathcal{T}^N V \le V^*.$$

Indeed, the right hand side is due to the maximizing property of optimal value function. The left hand side follows from standard computations

$$
\begin{aligned}
V^*(s) - [\mathcal{T}V](s) &= [\mathcal{T}V^*](s) - [\mathcal{T}V](s) \\
&= \max_a (r(s,a) + \gamma P(\cdot|s,a)^T V^*) - \max_a (r(s,a) + \gamma P(\cdot|s,a)^T V) \\
&\le \gamma \max_a |P(\cdot|s,a)^T (V^* - V)| \\
&\le \gamma \max_a \|P(\cdot|s,a)\|_1 \|V^* - V\|_\infty \\
&\le \gamma \|V^* - V\|_\infty.
\end{aligned}
$$

$\square$

## C.2 Overestimation errors

Consider the approximate value iteration scheme (AVI) with overestimation errors [14]

$$\text{(AVI)} \qquad V_{t+1} = \mathcal{T}V_t + \epsilon_{t+1},$$

where $\epsilon_t \in \mathbb{R}^{\mathcal{S}}$ is a vector of overestimation errors defined as a difference between applications of optimal Bellman operator to noisy and exact values

$$\epsilon_{t+1} := \mathcal{T}\hat{V}_t - \mathcal{T}V_t.$$

The above scheme with regularization results in the regularized approximate value iteration scheme (reg-AVI) with overestimation errors

$$\text{(reg-AVI)} \qquad V_{t+1} = \mathcal{T}_{\Omega_t} V_t + \bar{\epsilon}_{t+1},$$

where $\bar{\epsilon}_t \in \mathbb{R}^{\mathcal{S}}$ is a vector of overestimation errors defined as a difference between applications of optimal regularized Bellman operator to noisy and exact values

$$\bar{\epsilon}_{t+1} := \mathcal{T}_{\Omega_t}\hat{V}_t - \mathcal{T}_{\Omega_t} V_t.$$

## C.3 Error propagation

In the next lemma we extend the Lemma 5 to include errors $\bar{\epsilon}_t$ at computation of the regularized Bellman operator. The final bound in Proposition 2 is obtained by noticing that

$$\bar{\epsilon}_{t+1} := V_{t+1} - \mathcal{T}_{\Omega_t} V_t = V_{t+1} - \mathcal{T}_{\Omega_t} V_t + (\mathcal{T}V_t - \mathcal{T}V_t) = V_{t+1} - \mathcal{T}V_t - \Omega_t^*(A_{V_t}) = \epsilon_{t+1} - \Omega_t^*(A_{V_t}). \tag{27}$$

**Lemma 6** (Approximate regularized value function). *Consider the reg-AVI scheme with time-varying regularization function $\Omega_t(\pi) := \alpha_t \Omega(\pi)$, $\alpha_t > 0$, and per-iteration error vector $\bar{\epsilon}_t \in \mathbb{R}^{\mathcal{S}}$. Denote the approximate regularized value function after $N$ iterations $V_N := \mathcal{T}_{\Omega_N}(\mathcal{T}_{\Omega_{N-1}}(\dots(\mathcal{T}_{\Omega_1} V_0 + \bar{\epsilon}_1)\dots) + \bar{\epsilon}_{N-1}) + \bar{\epsilon}_N$. Then, $V_N$ satisfies*

$$\|V_N - V^*\|_\infty \le \sum_{t=1}^N \gamma^{N-t} (\|\bar{\epsilon}_t\|_\infty + \|\Omega_t^*(A_{V_{t-1}})\|_\infty) + \gamma^N \|V_0 - V^*\|_\infty.$$

*Proof.* The proof closely follows Lemma 5 with additional error vector. $\square$

**Proposition 2** (reg-AVI error propagation). *For any initial value function $V_0$, consider the entropy-based reg-AVI scheme (15) with time-varying regularization parameter $(\alpha_t)_t > 0$. Then, it holds that*

$$\|V_N - V^*\|_\infty \le E_N + A_N + \gamma^N \|V_0 - V^*\|_\infty, \tag{17}$$

*where $E_N := \sum_{t=1}^N \gamma^{N-t} \|\epsilon_t - \Omega_t^*(A_{V_{t-1}})\|_\infty$, $A_N := \sum_{t=1}^N \gamma^{N-t} \|\Omega_t^*(A_{V_{t-1}})\|_\infty$, the regularization gap $\Omega^*(A_V) := \mathcal{T}_\Omega V - \mathcal{T}V$ and the approximation errors of value update $\epsilon_t := V_t - \mathcal{T}V_{t-1}$.*

*Proof.* It is sufficient to invoke Lemma 6 with the error vector given by (27)

$$\bar{\epsilon}_t = \epsilon_t - \Omega_t^*(A_{V_{t-1}}).$$

$\square$

# D   Smooth regularized AVI

By combining smoothing with regularization, we obtain a smooth regularized Bellman operator. Similar to previous Section C, we first establish the distance between the smooth regularized value function to the optimal value function in Lemma 7.

## D.1   Smooth regularized value function

**Lemma 7** (Smooth regularized value function). *Consider the smooth reg-VI scheme with smoothing parameter $\beta \in [0, 1)$, time-varying regularization function $\Omega_t(\pi) := \alpha_t \Omega(\pi)$, $\alpha_t > 0$. Let $\tilde{\gamma} := \beta\gamma + (1 - \beta)$. Denote the smooth regularized value function after $N$ iterations $V_N := \mathcal{T}_{\Omega_N,\beta}(\mathcal{T}_{\Omega_{N-1},\beta}(\dots(\mathcal{T}_{\Omega_1,\beta}V_0)))$. Then, $V_N$ satisfies*

$$\|V_N - V^*\|_\infty \leq \beta \sum_{t=1}^{N} \tilde{\gamma}^{N-t}\|\Omega_t^*(A_{V_{t-1}})\|_\infty + \tilde{\gamma}^N\|V_0 - V^*\|_\infty.$$

*Proof.* Notice that Lemma 5 can be extended to the smooth regularized Bellman operators

$$\mathcal{T}_{\Omega,\beta}V = \beta(\mathcal{T}V + \Omega_t^*(A_V)) + (1 - \beta)V = \mathcal{T}_\beta V + \beta\Omega_t^*(A_V).$$

Similar to Lemma 5 and exchanging the optimal Bellman operator $\mathcal{T}$ to the smooth optimal Bellman optimal $\mathcal{T}_\beta$, it can be shown that

$$(\mathcal{T}_\beta V)^N \leq V_N \leq (\mathcal{T}_\beta V)^N + \sum_{t=1}^{N} \tilde{\gamma}^{N-t}\|\beta\Omega_t^*(A_{V_{t-1}})\|_\infty \mathbf{1}.$$

Using the same arguments as in Lemma 5, one has

$$V^* - \tilde{\gamma}^N\|V^* - V\|_\infty \mathbf{1} \leq (\mathcal{T}_\beta V)^N \leq V^*.$$

By combining the above bounds, one obtains the desired result. $\square$

## D.2   Error propagation of smooth regularized AVI

Consider the smooth reg-AVI scheme

$$V_{t+1} \leftarrow \mathcal{T}_{\Omega_t,\beta}V_t + \beta\bar{\epsilon}_t,$$

where $\bar{\epsilon}_t \in \mathbb{R}^\mathcal{S}$ is the reg-AVI error vector. Following the structure of Section C.3, we provide the distance between the smooth regularized approximate value function and the optimal value function in Lemma 8. The final bound is obtained in Theorem 1 by taking the reg-AVI error vector (27).

**Lemma 8** (Smooth regularized approximate value function). *Consider the approximate smooth reg-AVI scheme with smoothing parameter $\beta \in [0, 1)$, time-varying regularization function $\Omega_t(\pi) := \alpha_t \Omega(\pi)$, $\alpha_t > 0$ and per-iteration error vector $\bar{\epsilon}_t \in \mathbb{R}^\mathcal{S}$. Let $\tilde{\gamma} := \beta\gamma + (1 - \beta)$. Denote the smooth regularized approximate value function after $N$ iterations $V_N := \mathcal{T}_{\Omega_N,\beta}(\mathcal{T}_{\Omega_{N-1},\beta}(\dots(\mathcal{T}_{\Omega_1,\beta}V_0 + \beta\bar{\epsilon}_1)\dots) + \beta\bar{\epsilon}_{N-1}) + \beta\bar{\epsilon}_N$. Then, $V_N$ satisfies*

$$\|V_N - V^*\|_\infty \leq \beta\tilde{E}_N + \beta\tilde{A}_N + \tilde{\gamma}^N\|V_0 - V^*\|_\infty,$$

*where $\tilde{E}_N := \sum_{t=1}^{N} \tilde{\gamma}^{N-t}\|\bar{\epsilon}_t\|_\infty$, $\tilde{A}_N := \sum_{t=1}^{N} \tilde{\gamma}^{N-t}\|\Omega_t^*(A_{V_{t-1}})\|_\infty$.*

*Proof.* Similar to Lemma 6, the result follows by considering an additional vector of errors in Lemma 7. $\square$

**Theorem 1** (Smooth reg-AVI error propagation). *For any initial value function $V_0$, consider the smooth regularized AVI scheme (19) with smoothing parameter $\beta \in [0, 1)$ and time-varying temperature parameter $(\alpha_t)_t > 0$. Denote $\tilde{\gamma} := \beta\gamma + (1 - \beta)$. Then, one has*

$$\|V_N - V^*\|_\infty \leq \beta \tilde{E}_N + \beta \tilde{A}_N + \tilde{\gamma}^N \|V_0 - V^*\|_\infty, \qquad (20)$$

*where $\tilde{E}_N := \sum_{t=1}^{N} \tilde{\gamma}^{N-t} \|\epsilon_t - \Omega_t^*(A_{V_{t-1}})\|_\infty$, $\tilde{A}_N := \sum_{t=1}^{N} \tilde{\gamma}^{N-t} \|\Omega_t^*(A_{V_{t-1}})\|_\infty$, the regularization gap $\Omega^*(A_V) := \mathcal{T}_\Omega V - \mathcal{T}V$ and the approximation errors of value update $\epsilon_t := V_t - \mathcal{T}V_{t-1}$.*

*Proof.* It is sufficient to invoke Lemma 8 with the error vector given by (27)

$$\bar{\epsilon}_t = \epsilon_t - \Omega_t^*(A_{V_{t-1}}).$$

$\square$

## D.3 Neural network function approximation

Motivated by the Soft Actor-Critic algorithm, we consider the value network $V_\theta : \mathcal{S} \to \mathbb{R}$ trained using the smooth reg-AVI update and (continuous-time) gradient descent for optimization (see Section 5.3)

$$\text{(smooth reg-AVI)} \quad \theta_{k+1} \leftarrow \arg\min_\theta \|V_\theta - \mathcal{T}_{\Omega_k, \beta} V_k\|_2^2, \ V_{k+1} \leftarrow V_{\theta_{k+1}}. \qquad (28)$$

The central object of this study is the limiting *Neural Tangent Kernel (NTK)* [28] of the value network $V_\theta$, defined as

$$K(s, s') = \mathbb{E}_{\theta_0 \sim \text{init}} \left[ \frac{\partial V_\theta(s)}{\partial \theta}^T \frac{\partial V_\theta(s')}{\partial \theta} \Big|_{\theta = \theta_0} \right]. \qquad (29)$$

Below, we will show that the limiting NTK characterizes the approximation errors of (randomly initialized) value network with sufficiently large width trained using (continuous-time) gradient descent.

**Theorem 3** (Linear convergence of value network errors). *For notational simplicity, let us denote the value network $V(t) := V_\theta\big|_{\theta = \theta(t)}$ with sufficiently large width and the Bellman update $b_{k+1} := \mathcal{T}_{\Omega, \beta} V_k$. Then, if the limiting NTK of the neural net $V_\theta$ is positive definite, i.e its smallest eigenvalue is positive $\lambda_{\min}(K) > 0$, then the following contraction holds almost surely over all initializations $\theta(0)$ of the neural network*

$$|u_j^T(V(t) - b_{k+1})| \leq \exp(-\lambda_{\min}(K)t)|u_j^T(V(0) - b_{k+1})|, \ \forall t \geq 0, \ j \in [\![S]\!],$$

*where $u_1, \ldots, u_S$ are the eigenvectors of $K$. In particular, the process $V(t)$ converges (in any norm) to $b_{k+1}$ almost surely over random initializations of $V_\theta$ at linear rate $e^{-\lambda_{\min}(K)}$.*

*Proof.* Recall the dynamics of value function during the continuous gradient descent

$$\frac{dV_\theta}{dt} = \nabla_\theta V_\theta \dot{\theta}(t) = -K(t)(V_{\theta(t)} - \mathcal{T}_{\Omega, \beta} V_k)\big|_{\theta = \theta(t)}. \qquad (30)$$

It has been shown in [19] that $K(t) \overset{\text{a.s.}}{=} K + \mathcal{O}(m^{-1/2})$ for large $m := \min(m_1, \ldots, m_L)$, where $K$ is the limiting NTK given by (29).

Let $K = \sum_{j=1}^{S} \lambda_j u_j u_j^T$ be the eigenvalue decomposition of $K$, where $\lambda_{\max}(K) := \lambda_1 \geq \cdots \geq \lambda_S =: \lambda_{\min}(K) \geq 0$ are the eigenvalues and $u_1, \ldots, u_S$ are the eigenvectors, with $u_j^T u_k = 0$ for all $j, k \in [\![S]\!]$ such that $j \neq k$.

---

**Technical note.** Further suppose $\lambda_{\min}(K) > \sup_{t \geq 0} \|K(t) - K\|$. This assumption is not at all restrictive. For example, it is sufficient to have $\lambda_{\min}(K) = \Omega(1)$ (i.e the eigenvalues of $K$ are positive and bounded away from zero). This is because $\|K(t) - K\| = \mathcal{O}(m^{-1/2})$ as explained before. Then, we can use $K$ (the NTK) in place of $K(t)$ in the remainder of the analysis.

---

Multiplying both sides of (30) by the $j$th eigenvector $u_j$ of $K$, and simplifying, we get

$$\frac{d(u_j^T V(t))}{dt} = u_j^T \frac{dV(t)}{dt} = -u_j^T K(V(t) - b_{k+1}) = -\lambda_j u_j^T(V(t) - b_{k+1})),$$

Integrating the above equation w.r.t time $t$ and taking absolute values then gives for all $t \geq 0$,

$$|u_j^T(V(t) - b_{k+1})| = \exp(-\lambda_j t)|(u_j^T(V(0) - b_{k+1})|$$
$$\leq \exp(-\lambda_{\min}(K)t)(u_j^T|V(0) - b_{k+1}|).$$

Because the eigenvectors $u_1, u_2, \ldots$ are pairwise orthogonal, we have $\|V(t) - b_{k+1}\|_2 \leq \mathcal{O}(e^{-\lambda_{\min}(K)t})\|V(0) - b_{k+1}\|_2$, and so $V(t)$ converges to $b_{k+1}$ in the limit as $t \to \infty$ (in any norm, since all norms are equivalent in finite-dimensional spaces) exponentially fast with rate $e^{-\lambda_{\min}(K)}$. $\qquad\square$

**Example.** If $V_\theta$ is a two-layer ReLU network under a certain form, then it can be shown that the corresponding limiting NTK is given by $K(s, s') = \|s\|\|s'\|\kappa\left(\frac{s^T s}{\|s\|\|s'\|}\right)$, where $\kappa(u) := \frac{1}{\pi}(u(\pi - \arccos(u)) + \sqrt{1 - u^2})$.

**Corollary 2** (Value network errors of the smooth reg-AVI). *Under conditions of Theorem 2, the neural network approximation error of problem* (22) *optimized by gradient descent is upper bounded by*

$$\|\epsilon_{k+1}^a\| = \|V_{\theta_{k+1}} - \mathcal{T}_{\Omega,\beta} V_k\| \overset{a.s.}{=} \mathcal{O}(e^{-\lambda_{\min}(K)T}),$$

*where $K$ is the limiting NTK of the value network $V_\theta$ and $T$ is the number of gradient descent steps.*

*Proof.* The result follows from Theorem 3 at $t \to \infty$ due to the norm equivalence in finite dimensions. It holds in any norm at the price of changing the constants in the $\mathcal{O}(m^{-1/2})$ (this constant is $\sqrt{S}$ for the $\ell_\infty$-norm). As a consequence of Theorem 3, the statement should be understood in an almost-sure sense over random initializations of the neural network $V_\theta$. $\qquad\square$