[Reviews · NeurIPS 2020]

Review 1

Summary and Contributions: This work is a theoretical analysis of recently introduced reinforcement learning techniques. In particular, the work analyzes entropy regularization and action-value smoothing from an approximate dynamic programming point of view, investigating the error bounds that can be derived for the techniques included. The main claims are that (1) smoothing increases stability and that (2) entropy regularization reduces overestimation error. The work concludes with some comments on neural network function approximation in the infinite-width limit.

Strengths: - I have verified the proofs, and except for some things I note in the weaknesses, the work is technically sound. - As far as I know, the analysis is novel. The formulation of the error bounds in terms of the regularization gap is a useful concept, - The ideas are presented simply, in a manner clear and understandable for an RL audience. I find that it's quite difficult, although essential, to present work in a way as if you had yourself thought of it. In general, I think the field could benefit from being more accepting of simple papers that may not appear complicated, but look at important questions. - I appreciated that the theory was motivated with connections to observed phenomena in the literature. In particular, the increased stability resulting from smoothing, and the impact of regularization on overestimation bias.

Weaknesses: - While I understand the space limitations, I think the paper could greatly benefit from more explanation of the meaning of the bounds (perhaps in the appendix). - Line 122: it's not obvious to me that the smoothed bound is more stable, since the \gamma factor in the numerator is also larger. Some calculations here, or a very simple experiment, would greatly help the reader understand when smoothing would be desirable. - The above also applies for the discussion on overestimation starting on line 181, especially in the trade-off of reducing overestimation error and converging to a suboptimal value function. - The above applies for the combined smoothness + regularization algorithm

Correctness: - The claims are correct for the most part, excepting some questions I had about the neural network function approximation section. As this claim doesn't seem to be major, I am willing to weight it less and put the paper at an accept for now. - Explicitly including the approximations in eq (24) (i.e., O(m^{-1/2})) in the proof would help. I don't completely follow the argument given, since the use of limiting approximations doesn't seem to allow the use of any inequalities in lines 482-483. This could just be my relative unfamiliarity with NTK. - What is "overwhelming probability"? Is this almost surely? - How is the first inequality of lines 482-483 achieved? Where does the u_j go? How do we go from V(t) to V(0)? - The second product term after the second inequality of lines 482-483 seems to be missing a +1 from the other exp term in the previous line. - I'm also unsure how the integration wrt time gives us the equality right after line 482.

Clarity: Yes, excepting some comments I have made in Additional Feedback and in Weaknesses.

Relation to Prior Work: - Yes. The work contextualizes itself in the light of recent experimental developments in RL, and provides explicit differences from prior work (lines 46-49). - Missing work - Smoothing alg: http://proceedings.mlr.press/v97/lazic19a/lazic19a.pdf - entropy reg and optimization landscape: https://arxiv.org/abs/1811.11214

Reproducibility: Yes

Additional Feedback: - Typos - Line 67: Bellman --> Bellman operator - Line 108: approximation --> approximation optimal policies - Line 158-159: I believe this equation should be a lemma, as it is in the appendix - Line 362: \forall \pi' --> \forall \pi_N - Eq (26) is missing a \gamma term on the RHS - Eq (27) is missing a \tilde{\gamma} term on the RHS - between lines 406-407, (TV)^2 --> T^2 V; same for many expressions in Lemma 3 - Lines 412-413: should be the usual bellman operator instead of the regularized operator - Lines 414-415, 418-419: shouldn't the \hat be over the Bellman operators instead of over V? - I think it might be more clear to replace eq 15 with eq 28 in the appendix. I found the latter definition of the regularized errors clearer. - Confusing notation between regularization gap and regularization function. Different letters would be better ---------------------- I thank the authors for the response! All of my concerns have been addressed.


Review 2

Summary and Contributions: In this paper, the authors investigate the effect of the widely used technique in RL: the moving average target network, entropy regularization, and neural network parametrization. Based on the approximate dynamics programming framework, the error induced by these components are characterized, which leads to a better understanding of the reason of the practical success.

Strengths: This is a solid theoretical paper with important contribution to RL community. The error propagation analysis reveals the reasonable explanation to the widely used components in the successful practical RL algorithms.

Weaknesses: I have read the reply from the authors. Hope the derivation of Corollary 1 and Theorem 1 will be more clear in the final version. ========= I am quite pleasant to read this submission. However, the analysis for approximation error via NTK is not clear to me: what is the norm in Corollary 1 and how that is derived from Theorem 1 with the projection error? The approximation error characterization seems can be a straightforward application of Theorem 3.2 in [1]. [1] On Exact Computation with an Infinitely Wide Neural Net. Sanjeev Arora, Simon S. Du, Wei Hu, Zhiyuan Li, Ruslan Salakhutdinov, Ruosong Wang.

Correctness: As far as I can tell, the conclusions in the paper are reasonable.

Clarity: Yes. The logic flow of the paper is clear. It will be better if the approximation error part can be reduced by directly applying existing results, while leave the space for at least one proof of error propagation result.

Relation to Prior Work: Most of the important references are cited. It will be better if the advanced NTK result in [1] is cited.

Reproducibility: Yes

Additional Feedback:


Review 3

Summary and Contributions: This theory paper analyzes error propagation of basic components of modern deep reinforcement learning algorithms, particularly smoothed value functions, entropy regularization, and value function approximators. The analysis shows how value smoothing results in lower value error and thus improved stability at the cost of convergence speed, and that entropy regularization can mitigate positive value bias at the cost of optimizing a modified objective. The paper further bounds the value approximation error of neural network value in terms of the network width. --- Thank you for the response. I realize my earlier score did not fully capture the significance and quality of the work, and I have adjusted it accordingly (6->7).

Strengths: The paper nicely draws together results regarding value function error bounds and error propagation in a form that could be easily applied to practice. Even though the paper does not involve experimental section, it sheds light on how commonly used tricks to stabilize DRL algorithm work, and how they could be improved.

Weaknesses: I only have minor complaints. The definition of smoothed Bellman operator does not match what's typically used in practice (exponential average vs exponential Polyak average), and some analysis or comment on the difference would make the paper better.

Correctness: I did not check the proofs in detail, but I did not spot any obvious flaws.

Clarity: The paper is generally well written and easy to follow. Section 5.3. would have deserved a longer explanation. As a non-theorist, I found it too dense, and difficult to grasp its significance.

Relation to Prior Work: Prior work is covered only briefly, I'm not familiar enough with this topic to assess whether all relevant work is considered sufficiently.

Reproducibility: Yes

Additional Feedback:


Review 4

Summary and Contributions: The paper studies key components of popular RL algorithms like Soft Actor-Critic (SAC). It shows that value smoothing (exponential moving average in the target value during TD updates) increases training stability (Proposition 1 and Theorem 1), entropy regularization cancels out value overestimation (Proposition 2 and Theorem 1), and that the error of TD updates with neural net value functions is small (Theorem 2).

Strengths: 1. The paper explains the important techniques of value smoothing and entropy regularization, which have shown widespread success in practice but lack theoretical support. The proofs presented here can be significant to guiding developing practical algorithms. 2. The paper uses error propagation to study algorithms like SAC. This method appears less common in theoretic papers and could offer some new insights.

Weaknesses: 1. SAC uses exponential moving average on the target network weights, instead of on the predicted values as analyzed in the paper. This isn’t a major difference and could be argued away using Lipschitz constants, but it’s worth mentioning. 2. The goal of SAC and similar algorithms is to find the maximum entropy policy, not the optimal (unregularized) value V^* as analyzed in Theorem 1 and alike. This doesn’t impact the contributions of this paper, but is worth mentioning and analyzing. 3. For continuous actions, finding the Boltzmann policy (eqn (10) here, or eqn (10) in the SAC paper) a very challenging problem. Given that much attention in SAC and similar papers is devoted to solving it, I think it’s worth some discussion.

Correctness: I have read through the main text and part of the appendix. I am confident that the proofs are correct, but it is possible that I have missed some details.

Clarity: The paper is well written and easy to follow.

Relation to Prior Work: The paper clearly discusses its differences from prior works.

Reproducibility: Yes

Additional Feedback: I have read the authors’ feedback. I will keep my score. ------------------------------- Since the Universal Approximator Theorem already shows that the approximation error \eps^a is small, Theorem 2 feels less useful compared to other arguments made here.

[Author Response · NeurIPS 2020]

We thank the reviewers for their valuable feedback. In this rebuttal, we address the reviewers' comments and questions.

**[R1]** *"Line 122: it's not obvious to me that the smoothed bound is more stable"* The bound (8) stabilizes training by
multiplying the error term $E_N$ by a smoothing factor $\beta \in (0, 1]$ at a given initialization. If $\beta$ is sufficiently small, then
the bound is dominated by the initialization term that decreases at rate $\tilde{\gamma}$, slower than the standard AMPI scheme since
$\tilde{\gamma} \geq \gamma$. The bound (8) does not contain $\gamma$ in the numerator – that was a typo in paper that we corrected in the Appendix,
so the Eq. (26-27) are correct. The reason is that in the proof of Lemma 1, line 366 contains the loss at $N - 1$ step.

**[R1]** *"Very simple experiment, would greatly help the reader"* We include in this rebuttal a numerical illustration of
the smoothing technique (these results have also been integrated in the manuscript). We run experiments on a toy
stochastic gridworld problem with the evaluation step error due to the sampling of state-transitions. We plot the average
performance loss over 30 runs with varying values of smoothing factor $\beta$. As can be seen from the figure below, smaller
   values of $\beta$ result in tighter confidence intervals, but slower convergence speed.

**[R1]** *"The above also applies for the discussion on overestimation"*. The advantage of the bound (16) is that the term
$E_N$ could be made small by appropriately tuning the temperature parameter $\alpha_t$ to get the regularization gap $\Omega_t^*(A_{V_t})$ to
match the overestimation error $\epsilon_t$. In exchange, the $A_N$ term remains and results in a different regularized fixed point.

**[R1]** *"The above applies for the combined smoothness + regularization algorithm"* The combined bound (19) is
beneficial since (a) it downweights terms $\tilde{E}_N$ and $\tilde{A}_N$ by a factor of $\beta \in (0, 1]$, (b) allows to adjust the temperature
parameter $\alpha_t$ to match the noise level, making the term $\tilde{E}_N$ even smaller.

**[R1] On (24) and the approximation $K(t) = K + \mathcal{O}(m^{-1/2})$.** We use the standard NTK arguments: if we taylor-
expand $K(t)$ (defined in (24)) around the initialization parameters $\theta(0)$, then all terms of order 1 and higher are
$\mathcal{O}(m^{-1/2})$. Thus, $K(t) = K + \mathcal{O}(m^{-1/2})$, where $K = K(0)$. The use of $\mathcal{O}(\ldots)$ is in the following sense: the
constants are absolute (don't depend on $m$). This is because we suppose the initial NN parameters $\theta(0)$ are made of
random matrices with iid $\mathcal{N}(0, m^{-1})$ entries.

**[R1] On the proof of Theorem 2.** The disappearance of $u_j$ in the first inequality in lines $482 - 483$ is a typo; the
stray term $+e^{-\lambda_{\min}(K)}$ is a typo too; these will be corrected in the manuscript. We repeat the corrected argument
for lines $482 - 483$ here. Define $v_j(t) := u_j^T(V(t) - b_{k+1})$, where $V(t) := V_{\theta(t)}$ as usual. Then, the ODE just
before line 482 can be rewritten as $\frac{d}{dt}v_j(t) = -\lambda_j v_j(t)$. Integrating this ODE w.r.t time $t$ gives $v_j(t) = e^{-\lambda_j t}v_j(0)$
$\forall t \in \mathbb{R}$. Taking absolute values gives $|v_j(t)| = e^{-\lambda_j t}|v_j(0)| \leq e^{-\lambda_{\min}(K)t}|v_j(0)|$, and plugging-in the definition of
$v_j(t)$ above then gives $|u_j^T(V(t) - b_{k+1})| \leq e^{-\lambda_{\min}(K)t}|u_j^T(V(0) - b_{k+1})|$, $\forall j \in [\![S]\!]$, $t \in \mathbb{R}$, and large $m$. Because
the eigenvectors $u_1, u_2, \ldots$ are pairwise orthogonal, we deduce that $\|V(t) - b_{k+1}\|_2 \leq e^{-\lambda_{\min}(K)t}\|V(0) - b_{k+1}\|_2$,
and so $V(t)$ converges to $b_{k+1}$ (in any norm, since all norms are equivalent in finite-dim. spaces) exponentially fast.

**[R1]** *"Missing work"*. We agree that other perspectives on the entropy regularization are worth mentioning, such as the
ones you suggest on the loss landscape smoothing and value averaging effect of the KL divergence regularization.

**[R1]** *What is "overwhelming probability"?* Our use of this term was a misnomer. Indeed, those approximation
statements are to be understood in an *almost-sure* sense i.e $\mathbb{P}(\text{statement holds for large } m) = 1$, over all random
initializations of the parameters of the neural network $V_\theta$. We note that we might be able to have more quantitative
(nonasymptotic) statements using the results from [1], as suggested by **[R2]**.

**[R2]** *"What is the norm in Cor. 1 and how that is derived from Thm. 1 with the projection error?"* Cor. 1 bounds the
norm of the approximation error vector $\epsilon_{k+1}^a$ that appears in Thm. 1. Due to the norm equivalence in finite dimensions,
Cor. 1 holds in any norm (at the price of changing the constants in the $\mathcal{O}(m^{-1/2})$; this constant is $\sqrt{S}$ for the $\ell_\infty$-norm).
Cor. 1 follows from Thm. 2, and should be understood in an almost-sure sense: $\mathbb{P}(\|V_{\theta_{k+1}} - b_{k+1}\|_\infty = \mathcal{O}(m^{-1/2})) = 1$,
over random initializations of the neural net $V_\theta$.

**[R3, R4]** Indeed, we analyze the smoothing update on state values instead of weights. As R4 mentioned, this can be
argued away using Lipschitz constants. We will add a discussion about that.

[Meta-Review · NeurIPS 2020]

This paper analyzes through the lens of approximate dynamic programming the popular techniques of Q-values smoothing (with target networks) and entropy regularization. This analysis provides theoretical insights explaining their empirical success. After author feedback and discussion all reviewers agree that this is a meaningful contribution to the better understanding of existing RL algorithms. This is thus a clear « Accept » decision. That being said, I would like to ask the authors to please add a discussion w.r.t. recent closely related work "Leverage the Average: an Analysis of Regularization in RL", which is current missing from the references.